# Common Risk Factors in Relatives and Spouses of Patients with Type 2 Diabetes in Developing Prediabetes

**DOI:** 10.3390/healthcare9081010

**Published:** 2021-08-06

**Authors:** Wei-Hao Hsu, Chin-Wei Tseng, Yu-Ting Huang, Ching-Chao Liang, Mei-Yueh Lee, Szu-Chia Chen

**Affiliations:** 1Department of Internal Medicine, Kaohsiung Municipal Siaogang Hospital, Kaohsiung Medical University, Kaohsiung 812, Taiwan; my345677@yahoo.com.tw; 2Division of Endocrinology and Metabolism, Department of Internal Medicine, Kaohsiung Medical University Hospital, Kaohsiung Medical University, Kaohsiung 807, Taiwan; 930263@kmuh.org.tw; 3Department of Nursing, Kaohsiung Medical University Hospital, Kaohsiung 807, Taiwan; 4Statistical Analysis Laboratory, Department of Medical Research, Kaohsiung Medical University Hospital, Kaohsiung Medical University, Kaohsiung 807, Taiwan; stakmuh@gmail.com; 5Department of Laboratory Technology, Kaohsiung Municipal Siaogang Hospital, Kaohsiung 812, Taiwan; k670806@yahoo.com.tw; 6Faculty of Medicine, College of Medicine, Kaohsiung Medical University, Kaohsiung 807, Taiwan; 7Division of Nephrology, Department of Internal Medicine, Kaohsiung Medical University Hospital, Kaohsiung Medical University, Kaohsiung 807, Taiwan; 8Research Center for Environmental Medicine, Kaohsiung Medical University, Kaohsiung 807, Taiwan

**Keywords:** relatives, spouses, type 2 diabetes, family history, prediabetes

## Abstract

Prediabetes should be viewed as an increased risk for diabetes and cardiovascular disease. In this study, we investigated its prevalence among the relatives and spouses of patients with type 2 diabetes or risk factors for prediabetes, insulin resistance, and β-cell function. A total of 175 individuals were included and stratified into three groups: controls, and relatives and spouses of type 2 diabetic patients. We compared clinical characteristics consisting of a homeostatic model assessment for insulin resistance (HOMA-IR) and beta cell function (HOMA-β), a quantitative insulin sensitivity check index (QUICKI), and triglyceride glucose (TyG) index. After a multivariable linear regression analysis, the relative group was independently correlated with high fasting glucose, a high TyG index, and low β-cell function; the relatives and spouses were independently associated with a low QUICKI. The relatives and spouses equally had a higher prevalence of prediabetes. These study also indicated that the relatives had multiple factors predicting the development of diabetes mellitus, and that the spouses may share a number of common environmental factors associated with low insulin sensitivity.

## 1. Introduction

Prediabetes is typically defined as a higher blood glucose level than normal, but lower than the threshold for the criterion of diabetes [1]. The American Diabetes Association (ADA) defines prediabetes as a glycated hemoglobin (HbA1c) level of 5.7% to 6.4% or an impaired glucose tolerance (IGT) (2 h plasma glucose during an oral glucose tolerance test 140 to 199 mg/dL) or an impaired fasting glucose (IFG) level of 100 to 125 mg/dL [2]. The Nutrition and Health Survey in Taiwan 2013–2016 showed that 29.6% of adults in Taiwan have prediabetes defined by impaired fasting glucose alone [3]. The same factors associated with an increased risk of type 2 diabetes are also associated with an increased risk of prediabetes: overweight, large waist, diet (such as red meat, processed meat, and sugar-sweetened beverages), physical inactivity, old age, family history, race (such as African Americans, Hispanics, Native Americans, Asian Americans and Pacific Islanders), gestational diabetes, polycystic ovary syndrome, obstructive sleep apnea, and smoking tobacco [2,4]. A family history of diabetes has been shown to be a strong risk factor type 2 diabetes, which may at least partially be due to shared genetic and environmental factors [5,6], which was the finding of a U.S. population study [7]. This indicated that the risk of developing type 2 diabetes increased with the number of family members that had it. Prediabetes should be viewed as an increased risk for diabetes and cardiovascular disease (CVD) rather than a separate clinical entity [2,6]. In this study, we investigated the prevalence of prediabetes in the relatives and spouses of type 2 diabetes patients and the risk factors for prediabetes, insulin resistance, and β-cell function in these populations. These risk factors may result in the development of type 2 diabetes in family members including relatives and spouses.

## 2. Materials and Methods

### 2.1. Study Participants and Design

We enrolled 98 first-degree relatives (either or both parents, and siblings) who lived in a different household from that of the index type 2 diabetic patients and 37 spouses of patients with type 2 diabetes from the Endocrinology and Metabolism outpatient clinic of Kaohsiung Medical University Hospital, Kaohsiung City, Taiwan. Additionally, 40 healthy controls who had participated in a community screening program in 2019 and 2020 were enrolled, all of whom were aged ≥40 years and therefore eligible for a routine screening examinations every 3 years. The control subjects were excluded if they had a past history of prediabetes, diabetes, or a family history of diabetes mellitus. We attempted to invite all first-degree relatives and spouses to accompany the type 2 diabetic patients, but 2 relatives and 13 spouses refused to participate in the study for personal and geographic reasons. In addition, 8 healthy controls had lost contact with the clinic and 2 lacked information. This study was approved by the Institutional Review Board of Kaohsiung Medical University Hospital (KMUHIRB-F(I)-20180052), and all participants provided written in-formed consent, including for the publication of clinical details. In addition, all clinical investigations were carried out in accordance with the principles of the Declaration of Helsinki. 

### 2.2. Laboratory Investigations

An autoanalyzer (COBAS Integra 400 plus; Roche Diagnostics, www.roche.com/diagnostics/ (accessed on 23 June 2021) was used to measure biochemical parameters and urinary albumin and creatinine levels from 1-spot urine samples.

### 2.3. Definitions

#### 2.3.1. Albuminuria and Estimated Glomerular Filtration Rate (eGFR)

Estimated glomerular filtration rate (eGFR) was measured using the 4-variable Modification of Diet in Renal Disease (MDRD) Study equation. Albuminuria was defined as a urinary albumin–creatinine ratio ≥30 mg/g.

#### 2.3.2. Oral Glucose Tolerance Test (OGTT)

The OGTT is the gold standard diagnostic test for prediabetes and diabetes mellitus. The OGTT, which required overnight fasting, allowed the assessment of a glucose response after an oral glucose challenge and identified more individuals with dysglycemia compared to the fasting plasma glucose (FPG) or HbA1c. The preparation and administration of the OGTT was important for ensuring the validity of the test results. It included an assessment of both th FPG and the 2 h postprandial glucose (2 h PG) test after the oral 75 g glucose load. The results were interpreted based on venous plasma glucose levels before and 2 h after a 75 g oral glucose load with cut-off values of 140–199 mg/dL for prediabetes and >200 mg/dL for diabetes mellitus.

#### 2.3.3. Homeostatic Model Assessment for Insulin Resistance (HOMA-IR), Beta Cell Function (HOMA-β) and Quantitative Insulin Sensitivity Check Index (QUICKI)

HOMA-IR, HOMA-β and QUICKI are used to quantify the degrees of insulin resistance and β-cell secretory capacity. HOMA uses fasting measurements of blood glucose and insulin to calculate indices of insulin sensitivity and β-cell function. HOMA assumes that blood glucose and insulin concentrations are directly related due to an increase in insulin secretion through the effect of glucose on β-cells. HOMA-IR has been widely used as an index of insulin resistance in clinical and epidemiological studies. It is calculated as fasting insulin (mIU/L) × fasting glucose (mmol/L)/22.5. HOMA-β is calculated as: 20 × insulin (mIU/L)/(glucose (mmol/L) − 3.5), and QUICKI is derived using the inverse of the sum of the logarithms of fasting insulin and fasting glucose as: 1/(log(fasting insulin μU/mL) + log(fasting glucose mg/dL)).

#### 2.3.4. Triglyceride-Glucose (TyG) Index 

The TyG index is calculated as Ln (fasting triglycerides (mg/dL) × fasting blood glucose (mg/dL)/2).

### 2.4. Questionnaire

All of the participants completed a questionnaire to collect data on sociodemographics, personal and family histories of chronic diseases (diabetes mellitus, hypertension, and coronary artery disease), and employment. The spouses of the type 2 diabetic patients were also asked about marriage and cohabitation status. 

### 2.5. Statistical Analysis 

Data were expressed as median (25–75th percentile), mean ± standard deviation or percentage for β-cell function, HOMA-IR, triglycerides and insulin. The study patients were classified into three groups: (1) controls; (2) relatives; and (3) spouses of the patients with diabetes. Multiple inter-group comparisons were performed using one-way analysis of variance followed by a post hoc Bonferroni correction. Multivariable linear regression analysis was used to identify factors associated with fasting glucose, the OGTT, HbA1c, TyG index, HOMA-IR, β-cell function and QUICKI after adjusting for group, age, sex, hypertension, coronary artery disease, body mass index (BMI), fasting glucose, log-transformed triglycerides, total cholesterol, high-density lipoprotein–cholesterol, eGFR, GOT, GPT and a urinary albumin–creatinine ratio >30 mg/g. The control group was treated as the reference group. A difference was considered significant at *p* < 0.05. All statistical analyses were performed using SPSS version 19.0 for Windows (SPSS Inc. Chicago, IL, USA).

## 3. Results

A total of 175 individuals were included (60 men and 115 women, mean age 55.6 ± 11.4 years) and stratified into three groups: (1) controls (*n* = 40); (2) relatives (*n* = 98); and (3) spouses (*n* = 37). A comparison of the clinical characteristics of these study groups is shown in Table 1.

Compared to the controls, the relatives were older and predominantly female, They had more prescription medications for chronic diseases and a higher prevalence of hypertension, impaired glucose tolerance and fasting glucose, and prediabetes (HbA1c, 5.7–6.4%). They also had a higher BMI, fasting glucose, OGTT and HbA1c. In addition, the relatives had a lower QUICKI. Compared to the controls, the spouses were older, had more prescription medications for chronic diseases and a higher prevalence of hypertension, impaired glucose tolerance, and prediabetes (HbA1c, 5.7–6.4%) as well as a higher OGTT and HbA1c. 

### 3.1. Determinants of Fasting Glucose 

Table 2 shows the determinants of fasting glucose in all of the study participants. In the multivariable linear regression analysis––after adjusting for group, age, sex, hypertension, coronary artery disease, BMI, fasting glucose, log-transformed triglycerides, total cholesterol, high-density lipoprotein–cholesterol, eGFR, GOT, GPT and a urinary albumin–creatinine ratio >30 mg/g––old age, and a high BMI were independently correlated with high fasting glucose for the relatives of a patient with type 2 diabetes (vs. controls; unstandardized coefficient β: 5.611; 95% confidence interval (CI), 1.282–9.940; *p* = 0.011).

### 3.2. Determinants of TyG Index

Table 3 shows the determinants of the TyG index for all of the study participants. After the multivariable linear regression analysis old age, a high BMI, and a high triglyceride level were independently associated with a high TyG index for the relatives of a patient with type 2 diabetes (vs. controls; unstandardized coefficient β: 0.053; 95% CI, 0.0132–0.093; *p* = 0.010).

### 3.3. Determinants of β-Cell Function

Table 4 shows the determinants of β-cell function for all of the study participants. After the multivariable linear regression analysis, a low GPT was independently associated with low β-cell function for relatives of a patient with type 2 diabetes (vs. controls; unstandardized coefficient β: −43.083; 95% CI, −83.235 to −2.9313; *p* = 0.036).

### 3.4. Determinants of QUICKI

Table 5 shows the determinants of QUICKI in all of the study participants. After the multivariable linear regression analysis was conducted, for the relative (vs. controls; unstandardized coefficient β: −0.017; 95% CI, −0.033 to −0.002; *p* = 0.027) or spouse (vs. controls; unstandardized coefficient β: −0.023; 95% CI, −0.042 to −0.004; *p* = 0.020) of a patient with type 2 diabetes, young age, and a high BMI were independently associated with a low QUICKI.

### 3.5. Age, Sex and BMI Matched Subgroup Analysis of “Relatives vs. Control”

Table 6 shows the comparison of clinical characteristics between controls and relatives of DM patients by age, sex and BMI match. Compared to the controls, the relatives had a higher prevalence of hypertension, higher fasting glucose, higher OGTT, higher HbA1c and higher eGFR. 

### 3.6. Determinants of Fasting Glucose, TyG Index, β-Cell Function Abd QUICKI in Control vs. Relatives by Age, Sex and BMI Match

Table 7 shows the determinants of fasting glucose, TyG index, β-cell function and QUICKI in control vs. relatives by age, sex and BMI match. After a multivariable linear regression analysis, being a relative of a patient with type 2 diabetes (vs. controls) was significantly associated with a high fasting glucose (unstandardized coefficient β: 6.600; 95% CI, 2.245 to 10.774; *p* = 0.002) and a high TyG index (unstandardized coefficient β: 0.064; 95% CI, 0.024 to 0.104; *p* = 0.002). However, the relatives were not associated with β-cell function (*p* = 0.246) or QUICKI (*p* = 0.216).

### 3.7. Age, Sex and BMI Matched Subgroup Analysis of Spouse vs. Control

Table 8 shows the comparison of clinical characteristics between controls and spouses of DM patients by age, sex and BMI match. Compared to the controls, the spouses had a higher QUICKI. 

### 3.8. Determinants of Fasting Glucose, TyG Index, β-Cell Function Abd QUICKI in Control vs. Spouses by Age, Sex and BMI Match

Table 9 shows the determinants of fasting glucose, TyG index, β-cell function and QUICKI in control vs. spouses by age, sex and BMI match. After multivariable linear regression analysis, being the spouses of a patient with type 2 diabetes (vs. controls) were not associated with all parameters including fasting glucose (*p* = 0.994), TyG index (*p* = 0.975), β-cell function (*p* = 0.606) and QUICKI (*p* = 0.083).

## 4. Discussion

In this study, the relatives and spouses of patients with type 2 diabetes had a higher prevalence of prediabetes (55.1% and 54.1%, respectively) compared to the healthy controls (35.0%). In addition, they had a higher prevalence of IFG (56.1% and 59.5%, respectively) compared to adults in the 2005–2008 Nutrition and Health Survey in Taiwan (35.8%) [7]. Moreover, the relatives had a higher BMI than the healthy controls. These findings may explain the high prevalence of prediabetes among the relatives and spouses as previous studies have reported that old age, physical inactivity and being overweight are risk factors [2,4].

There are several important findings to this study. First, for the relatives, old age and a high BMI were associated with high fasting blood glucose. This was consistent with a previous study conducted in Taiwan in which old age and a high BMI corresponding to being overweight or obese were significantly associated with IFG [8]. In two nationwide cohorts of obese children and adolescents in Germany and Sweden, the risk of IFG was shown to be positively correlated with age and degree of obesity [9]. In addition, an epidemiological study conducted in Mexico reported that a family history of diabetes was associated with IFG, independent of BMI and age [10]. Our results also suggested that being a relative of a type 2 diabetic patient was a determinant of fasting glucose level.

Second, old age, a high BMI, and a high triglyceride level were independently associated with a high TyG index, which includes both fasting triglycerides and fasting glucose. It has been demonstrated to be a reliable marker of insulin resistance and developing diabetes [11,12]. A previous cross-sectional observational study including healthy non-diabetic young male adults showed that those with a family history of type 2 diabetes had a higher degree of insulin resistance [13]. Another report demonstrated that the risk of diabetes according to a family history could be classified as high, moderate, and average according to at least two generations, one generation, and no first-degree relatives with diabetes, respectively [14]. In addition, the degree of insulin resistance revealed a significant trend among the three risk categories, with the highest degree of insulin resistance in those with a high family history risk category of diabetes [14]. A short-term experimental study of healthy individuals with and without a family history of type 2 diabetes who were overfed by 5200 kJ/day for 28 days, showed that those with a family history of type 2 diabetes had greater insulin resistance by the end [15]. Aging was also associated with insulin resistance in several studies, with age-related decreased muscle mass, increased visceral fat deposition, less physical activity, and senile skeletal muscle dysfunction all contributing to insulin resistance [16,17,18,19]. A study also showed the BMI to be an independent predictor of insulin resistance [20]. In a cross-sectional study by González-Jiménez et al., participants who had abnormal values of HOMA-IR had a significantly higher BMI, indicating that excess body weight is an important predictor of insulin resistance [21]. Our results showed that being the relative of a patient with type 2 diabetes, old age and a high BMI were determinants of TyG index. This finding was consistent with previous studies showing that a family history of type 2 diabetes, aging and a high BMI are all associated with insulin resistance. 

Third, being a relative and having a low GPT level were independently associated with low β-cell function. In this study, we used HOMA-β to estimate beta cell function. HOMA-β has been shown to be moderately correlated with insulin secretion measured using hyperglycemic clamps, continuous infusion of glucose with model assessments, and acute insulin response estimated using the intravenous glucose tolerance test in both individuals with and without diabetes [22,23]. Stadler et al. reported that the first-degree offspring of a type 2 diabetic demonstrated insulin resistance and beta cell dysfunction in response to an oral glucose challenge [24]. In another study including normoglycemic male subjects of Hispanic origin, significantly lower HOMA-β values were noted in those with a family history of type 2 diabetes [25]. Our result showed that the relatives of type 2 diabetic patients were associated with a decline in β-cell function, which was consistent with these previous studies [24,25]. Several cross-sectional studies and observational cohort studies reported positive correlations between elevated ALT (GPT) and both systemic and hepatic insulin resistance, and that this could predict the development of type 2 diabetes [26,27]. A study of young obese patients reported that elevated ALT (GPT) and GGT levels were significantly associated with a decline in pancreatic islet β-cell function [28]. High plasma glucose levels induce toxicity and activate the apoptosis pathway in the liver [29]. These previous reports provided evidence that an elevated ALT (GPT) is significantly associated with insulin resistance and a decline in β-cell function, which was different to our findings. One possible explanation for this difference is that serum ALT (GPT) levels may have different roles for different age groups. In one retrospective cohort study, there was a linear association between serum ALT levels and all-cause mortality in adults aged <60 years, while the association was U-shaped in adults aged 60 and >60 years [30]. Serum ALT levels may have a similar relationship with β-cell function; however, further studies are needed to investigate the association. 

Fourth, for the relatives and spouses, young age, and a high BMI were independently associated with a low QUICKI, which was derived empirically through the mathematical transformation of plasma insulin and fasting blood glucose levels. It has been shown to be a reproducible, reliable, accurate and validated index of insulin sensitivity with good positive predictive power [31]. In addition, it has been shown to have a significantly better linear correlation with indices of insulin sensitivity in glucose clamp studies than in a minimal model or HOMA-IR [32]. HOMA-B is an indirect measure of beta cell function and only takes into account fasting plasma glucose and insulin level. HOMA yields limited information about intra-daily glucose fluctuations, and the model cannot accurately predict the impact of antidiabetic agents like insulin and insulin secretagogues on either beta cell function or tissue insulin sensitivity. Relatively low precision has been reported for estimates based on the HOMA model (~32% for HOMA-B; ~31% for HOMA-IR) [22]. More importantly, when plasma glucose levels are <63 mg/dL or ≤3.5 mmol/L, HOMA estimates cannot be used to assess beta cell function because they yield undefined or negative values. Furthermore, the interpretation of results on long duration type 2 diabetes mellitus when low fasting insulin was ≤5 μU/mL and fasting glucose was <81 mg/dL or 4.5 mmol/L was not valid. Caution is necessary when comparing HOMA values across different ethnicities, because the prevailing “normal” will vary based on genetics and the environment [33]. Moreover, QUICKI is a simple, useful, and inexpensive tool to measure insulin sensitivity that may be used effectively in large epidemiological or clinical research studies [32]. A study reported that relatives of patients with diabetes and a high BMI were associated with insulin resistance, and that this was consistent with the independent association of being the relative of a patient with diabetes, a high BMI, and low QUICKI (low insulin sensitivity). 

Insulin sensitivity is also affected by the amount of adipose tissue. The relationship between insulin resistance and visceral adipose tissue mass is directly proportional, and weight loss has been reported to improve insulin sensitivity. Hence, adipose tissue regulates insulin sensitivity in target tissues [34]. In addition, another study reported that he spouses of patients were also at a significantly higher risk of type 2 diabetes and glucose intolerance [35]. A systemic review and meta-analysis also suggested that the spouses were associated with a 26% increase in the risk of diabetes [36]. In the current study, we found that the spouses were significantly associated with low insulin sensitivity, and that this contributed to an increased risk of developing type 2 diabetes. A previous study found that the spouses of diabetic patients had a significantly higher BMI compared to spouses of individuals who did not have diabetes, and that the risk of diabetes in the spouses of patients with diabetes remained strong after adjusting for BMI [35]. We also found that being the spouse of a patient with diabetes was an independent determinant of low insulin sensitivity, independent of the BMI. A possible explanation for the increased risk of diabetes among spouses of diabetic patients is similarities in cigarette smoking, alcohol consumption, dietary habits, and physical activity [37,38]. We also found that young age was independently associated with a low QUICKI, which was different from previous studies. 

Two confounding factors may explain the difference. Exercise has been shown to be a valuable primary care strategy for healthy adults to improve insulin sensitivity, and short-term exercise has been shown to improve both insulin resistance and β-cell function in older people with impaired glucose tolerance [39,40]. In addition, regular physical activity has been shown to play an important role in the prevention and control of insulin resistance [41]. Therefore, physical activity is an important factor influencing insulin sensitivity. Both insulin resistance and an increased risk of type 2 diabetes have also been associated with smoking tobacco, which may be due to the reported association between nicotine and reduced insulin sensitivity [42,43]. In this study, we did not evaluate physical activity or cigarette smoking, and this may have influenced our results with regards to the relationship between age and insulin sensitivity. 

For our study to have a valuable and reliable conclusion, we need to performed further analysis of “relatives vs. control” and “spouse vs. control” subgroups concerning age, sex and BMI. The relatives were persistently associated with high fasting glucose and a high TyG index, but not with β-cell function and QUICKI. However, there was no greater association with fasting glucose, TyG index, β-cell function or QUICKI among the spouses. The study’s results might be due to the limited size of the study population, but we still believe they contribute very important information on, and have clinical significance for, prediabetes. In our future research, a larger study population will be enrolled.

There were other limitations to this retrospective study. First, we could not evaluate cause-and-effect relationships due to the cross-sectional study design. Second, we did not collect all possibly related sociodemographic data such as physical activity, dietary habits or cigarette smoking, and this may have influenced the associations among the study parameters. Third, we did not consider medications such as steroids which could have affected insulin resistance and β-cell function. Therefore, we could not adjust for the effect of these medications. Fourth, although the age of our study group was older than that for mature onset diabetes of the young (MODY) and latent autoimmune diabetes in adults (LADA), the genetic testing for MODY and autoantibody testing for LADA were not done, so we could not totally exclude the possibility that these patient groups might have been in our study population.

## 5. Conclusions

The relatives and spouses of type 2 diabetic patients in this study had a high prevalence of prediabetes. Being a relative was significantly associated with high fasting blood glucose, a high TyG index, low β-cell function, and low QUICKI, whereas being the spouse of a type 2 diabetic patient was only significantly associated with low QUICKI (Table 10). These findings indicated that the relatives had multiple factors predicting the development of diabetes mellitus, and that the spouses may share a number of common environmental factors associated with low insulin sensitivity. However, after being matched by age, sex and BMI with the control group, a relative was only significantly associated with high fasting blood glucose and the TyG index. These factors may explain the high prevalence of prediabetes in relatives and spouses of type 2 diabetic patients.

## Figures and Tables

**Table 1 healthcare-09-01010-t001:** Comparison of clinical characteristics between controls, relatives and spouses of DM patients.

Characteristics	Controls (*n* = 40)	Relatives (*n* = 98)	Spouses (*n* = 37)	*p*
Age (year)	48.2 ± 6.5	55.8 ± 11.9 *	62.9 ± 9.5 *^,†^	<0.001 *
Male gender (%)	50.0	28.6 *	32.4	0.053
Hypertension (%)	2.5	25.5 *	27.0 *	0.006 *
Coronary artery disease (%)	2.5	7.1	2.7	0.401
Chronic disease medication control (%)	7.5	53.9 *	52.0 *	<0.001 *
Waist circumference (cm)	79.9 ± 9.7	83.9 ± 13.1	84.7 ± 11.5	0.149
Body mass index (kg/m^2^)	23.1 ± 2.9	25.4 ± 5.2 *	24.3 ± 5.4	0.035 *
Laboratory parameters				
Fasting glucose (mg/dL)	96.9 ± 8.5	105.6 ± 12.8 *	102.2 ± 10.8	<0.001 *
Fasting glucose (mg/dL)				
<100 (%)	60.0	35.7 *	35.1	
100–125 (%)	40.0	56.1	59.5	0.068
≥126 (%)	0	7.1	5.4	
OGTT (mg/dL)	124.8 ± 29.1	153.9 ± 44.1 *	153.8 ± 51.8 *	0.001 *
OGTT (mg/dL)				
<140 (%)	72.5	46.9 *	45.9*	
140–200 (%)	27.5	39.8	40.5	0.032 *
≥200 (%)	0	13.3	13.5	
HbA_1c_ (%)	5.5 ± 0.4	5.8 ± 0.6 *	5.9 ± 0.6 *	0.001 *
HbA1c (%)				
<5.7 (%)	65.0	38.8 *	35.1 *	
5.7–6.4 (%)	35.0	55.1	54.1	0.020 *
≥6.5 (%)	0	6.1	10.8	
Triglyceride (mg/dL)	90.5 (57.3–112.3)	92.5 (64–152.8)	96 (66.5–128)	0.380
TyG index	8.3 ± 0.6	8.6 ± 0.6	8.5 ± 0.5	0.131
Total cholesterol (mg/dL)	209.1 ± 47.4	208.2 ± 43.4	212.6 ± 43.8	0.878
HDL-cholesterol (mg/dL)	61.2 ± 18.0	59.6 ± 17.9	63.6 ± 22.6	0.548
eGFR (mL/min/1.73 m^2^)	82.9 ± 9.3	87.6 ± 24.4	75.4 ± 15.8 ^†^	0.008 *
GOT (U/L)	25.4 ± 17.1	26.4 ± 12.5	25.4 ± 8.2	0.876
GPT (U/L)	26.8 ± 20.2	28.1 ± 17.4	28.6 ± 16.6	0.897
GA (%)	14.4 ± 1.9	14.8 ± 1.9	14.9 ± 1.8	0.377
Insulin (mU/L)	5.3 (3.2–9.8)	6.3 (4.2–9.1)	6.4 (4.5–9.1)	0.483
HOMA-IR	1.2 (0.8–2.4)	1.7 (1.1–2.4)	1.6 (1.1–2.3)	0.228
β-cell function	53.7 (36.0–108.1)	56.3 (39.2–79.3)	58.8 (42.3–89.6)	0.762
QUICKI	0.37 ± 0.06	0.36 ± 0.03 *	0.36 ± 0.03	0.033 *
U_ACR_ > 30 mg/g	15.0	8.2	10.8	0.496

Abbreviations. OGTT, oral glucose tolerance test; HbA_1c_, glycated hemoglobin; TyG, triglyceride-glucose; HDL, high-density lipoprotein; eGFR, estimated glomerular filtration rate; GOT, aspartate aminotransferase; GPT, alanine aminotransferase; GA, glycated albumin; HOMA-IR, homeostatic model assessment-insulin resistance; QUICKI, quantitative insulin sensitivity check index; U_ACR_, Urine albumin-to-creatinine ratio. * *p* < 0.05 compared with controls; ^†^
*p* < 0.05 compared with relatives of DM patients.

**Table 2 healthcare-09-01010-t002:** Determinants of fasting glucose using multivariable linear regression analysis.

Parameter	Multivariable	
	Unstandardized Coefficient β (95% CI)	*p*
Group		
Controls	Reference	
Relatives	5.611 (1.282, 9.940)	0.011 *
Spouses	1.789 (−3.626, 7.205)	0.515
Age (per 1 year)	0.208 (0.024, 0.391)	0.027 *
Male (vs. female)	3.099 (−0.751, 6.949)	0.114
Hypertension	0.055 (−4.387, 4.497)	0.981
Coronary artery disease	0.927 (−6.712, 8.567)	0.811
Body mass index (per 1 kg/m^2^)	0.619 (0.251, 0.987)	0.001 *
Triglyceride (log per 1 mg/dL)	5.674 (−3.587, 14.935)	0.228
Total cholesterol (per 1 mg/dL)	0.034 (−0.014, 0.081)	0.161
HDL-cholesterol (per 1 mg/dL)	−0.082 (−0.205, 0.041)	0.190
eGFR (per 1 mL/min/1.73 m^2^)	0.024 (−0.062, 0.110)	0.582
GOT (per 1 U/L)	−0.143 (−0.376, 0.090)	0.227
GPT (per 1 U/L)	0.115 (−0.060, 0.289)	0.196
U_ACR_ > 30 mg/g	−0.503 (−5.797, 4.791)	0.851

Values expressed as unstandardized coefficient β and 95% confidence interval (CI). Abbreviations are the same as in Table 1. * *p* < 0.05.

**Table 3 healthcare-09-01010-t003:** Determinants of TyG index using multivariable linear regression analysis.

Parameter	Multivariable	
	Unstandardized Coefficient β (95% CI)	*p*
Group		
Controls	Reference	
Relatives	0.053 (0.013, 0.093)	0.010 *
Spouses	0.016 (−0.034, 0.066)	0.519
Age (per 1 year)	0.002 (0.000, 0.004)	0.012 *
Male (vs. female)	0.029 (−0.007, 0.064)	0.113
Hypertension	0.002 (−0.039, 0.042)	0.941
Coronary artery disease	0.008 (−0.062, 0.079)	0.817
Body mass index (per 1 kg/m^2^)	0.005 (0.002, 0.009)	0.003 *
Triglyceride (log per 1 mg/dL)	2.356 (2.271, 2.442)	<0.001 *
Total cholesterol (per 1 mg/dL)	0.000 (0.000, 0.001)	0.101
HDL-cholesterol (per 1 mg/dL)	0.000 (−0.002, 0.000)	0.140
eGFR (per 1 mL/min/1.73 m^2^)	0.000 (0.000, 0.001)	0.493
GOT (per 1 U/L)	−0.001 (−0.004, 0.001)	0.174
GPT (per 1 U/L)	0.001 (0.000, 0.003)	0.158
U_ACR_ > 30 mg/g	−0.008 (−0.056, 0.041)	0.756

Values expressed as unstandardized coefficient β and 95% confidence interval (CI). Abbreviations are the same as in Table 1. * *p* < 0.05.

**Table 4 healthcare-09-01010-t004:** Determinants of β-cell function using multivariable linear regression analysis.

Parameter	Multivariable	
	Unstandardized Coefficient β (95% CI)	*p*
Group		
Controls	Reference	
Relatives	−43.083 (−83.235, −2.931)	0.036 *
Spouses	−33.939 (−84.171, 16.293)	0.184
Age (per 1 year)	−1.514 (−3.217, 0.188)	0.081
Male (vs. female)	−14.111 (−49.823, 21.601)	0.436
Hypertension	6.949 (−34.255, 48.154)	0.739
Coronary artery disease	16.098 (−54.760, 86.956)	0.654
Body mass index (per 1 kg/m^2^)	3.059 (−0.354, 6.472)	0.079
Triglyceride (log per 1 mg/dL)	−1.841 (−87.739, 84.058)	0.966
Total cholesterol (per 1 mg/dL)	0.055 (−0.385, 0.495)	0.805
HDL-cholesterol (per 1 mg/dL)	0.526 (−0.616, 1.668)	0.365
eGFR (per 1 mL/min/1.73 m^2^)	−0.243 (−1.043, 0.556)	0.549
GOT (per 1 U/L)	−1.454 (−3.612, 0.705)	0.185
GPT (per 1 U/L)	1.945 (0.329, 3.561)	0.019 *
U_ACR_ > 30 mg/g	16.529 (−32.578, 65.635)	0.507

Values expressed as unstandardized coefficient β and 95% confidence interval (CI). Abbreviations are the same as in Table 1. * *p* < 0.05.

**Table 5 healthcare-09-01010-t005:** Determinants of QUICKI using multivariable linear regression analysis.

Parameter	Multivariable	
	Unstandardized Coefficient β (95% CI)	*p*
Group		
Controls	Reference	
Relatives	−0.017 (−0.033, −0.002)	0.027 *
Spouses	−0.023 (−0.042, −0.004)	0.020 *
Age (per 1 year)	0.001 (0.000, 0.001)	0.032 *
Male (vs. female)	−0.006 (−0.019, 0.008)	0.404
Hypertension	−0.010 (−0.026, 0.006)	0.217
Coronary artery disease	−0.016 (−0.044, 0.011)	0.231
Body mass index (per 1 kg/m^2^)	−0.002 (−0.004, −0.001)	<0.001 *
Triglyceride (log per 1 mg/dL)	−0.015 (−0.048, 0.018)	0.368
Total cholesterol (per 1 mg/dL)	−5.678 × 10^−5^ (0.000, 0.000)	0.506
HDL-cholesterol (per 1 mg/dL)	−6.489 × 10^−5^ (0.000, 0.000)	0.769
eGFR (per 1 mL/min/1.73 m^2^)	9.310 × 10^−5^ (0.000, 0.000)	0.548
GOT (per 1 U/L)	0.000 (0.000, 0.001)	0.555
GPT (per 1 U/L)	0.000 (−0.001, 0.000)	0.087
U_ACR_ > 30 mg/g	−0.011 (−0.030, 0.008)	0.244

Values expressed as unstandardized coefficient β and 95% confidence interval (CI). Abbreviations are the same as in Table 1. * *p* < 0.05.

**Table 6 healthcare-09-01010-t006:** Comparison of clinical characteristics between controls and relatives of DM patients by age, sex and BMI match.

Characteristics	Controls (*n* = 39)	Relatives (*n* = 39)	*p*
Age (year)	48.2 ± 6.6	52.3 ± 11.6	0.060
Male gender (%)	48.7	41.0	0.495
Hypertension (%)	2.6	18.0	0.025 *
Coronary artery disease (%)	2.6	2.6	1.000
Waist circumference (cm)	79.8 ± 9.8	81.2 ± 10.1	0.558
Body mass index (kg/m^2^)	23.1 ± 2.9	23.7 ± 2.7	0.363
Laboratory parameters			
Fasting glucose (mg/dL)	96.8 ± 8.5	104.5 ± 9.9	0.001 *
Fasting glucose (mg/dL)			0.019 *
<100 (%)	61.5	30.7	
100–125 (%)	38.5	66.7	
≥126 (%)	0	2.6	
OGTT (mg/dL)	124.9 ± 29.5	149.7 ± 39.3	0.002 *
OGTT (mg/dL)			0.051
<140 (%)	71.8	51.3	
140–200 (%)	28.2	38.5	
≥200 (%)	0	10.3	
HbA_1c_ (%)	5.5 ± 0.4	5.8 ± 0.5	0.003 *
HbA1c (%)			0.058
<5.7 (%)	66.7	41.0	
5.7–6.4 (%)	33.3	56.4	
≥6.5 (%)	0	2.6	
Triglyceride (mg/dL)	102.3 (80.6–123.9)	112.5 (91.8–133.1)	0.492
TyG index	8.3 ± 0.6	8.5 ± 0.6	0.167
Total cholesterol (mg/dL)	210.3 ± 47.4	209.1 ± 39.1	0.901
HDL-cholesterol (mg/dL)	61.8 ± 17.8	59.2 ± 18.3	0.519
eGFR (mL/min/1.73 m^2^)	83.1 ± 9.4	91.5 ± 20.6	0.023 *
GOT (U/L)	25.5 ± 17.3	25.4 ± 11.9	0.988
GPT (U/L)	26.8 ± 20.5	26.9 ± 18.4	0.977
GA (%)	14.4 ± 1.9	14.9 ± 1.9	0.232
Insulin (mU/L)	10.5 (5.3–15.6)	7.3 (5.3–9.3)	0.241
HOMA-IR	2.5 (1.2–3.9)	1.8 (1.4–2.3)	0.307
β-cell function	118.2 (64.5–171.8)	71.8 (41.7–101.9)	0.131
QUICKI	0.38 ± 0.06	0.36 ± 0.03	0.180
U_ACR_ > 30 mg/g	15.4	5.1	0.136

Abbreviations are the same as Table 1. * *p* < 0.05.

**Table 7 healthcare-09-01010-t007:** Determinants of various parameters using multivariable linear regression analysis in control vs. relatives by age, sex and BMI match.

Parameter	Multivariable	
	Unstandardized Coefficient β (95% CI)	*p*
Fasting glucose		
Controls	Reference	
Relatives	6.600 (2.425, 10.774)	0.002 *
TyG index		
Controls	Reference	
Relatives	0.064 (0.024, 0.104)	0.002 *
β-cell function		
Controls	Reference	
Relatives	−38.502 (−104.191, 27.188)	0.246
QUICKI		
Controls	Reference	
Relatives	−0.015 (−0.039, 0.009)	0.216

Values expressed as unstandardized coefficient β and 95% confidence interval (CI). Abbreviations are the same as in Table 1. Adjusted for hypertension, coronary artery disease, body mass index, log triglyceride, total cholesterol, HDL-cholesterol, eGFR, GOT, GPT and U_ACR_ > 30 mg/g. * *p* < 0.05.

**Table 8 healthcare-09-01010-t008:** Comparison of clinical characteristics between controls and spouses of DM patients by age, sex and BMI match.

Characteristics	Controls (*n* = 13)	Relatives (*n* = 13)	*p*
Age (year)	53.9 ± 8.1	55.2 ± 8.7	0.695
Male sex (%)	31	15	0.352
Hypertension (%)	0	7.7	0.308
Coronary artery disease (%)	7.7	7.7	1.000
Waist circumference (cm)	78.8 ± 9.1	80.8 ± 14.1	0.659
Body mass index (kg/m^2^)	22.7 ± 3.2	21.4 ± 7.4	0.568
Laboratory parameters			
Fasting glucose (mg/dL)	95.2 ± 7.0	98.2 ± 10.3	0.384
Fasting glucose (mg/dL)			0.216
100–125 (%)	77.0	54.0	
≥126 (%)	23.0	46.0	
OGTT (mg/dL)	133.8 ± 31.0	141.48 ± 36.4	0.575
OGTT (mg/dL)			0.587
<140 (%)	62.0	53.9	
140–200 (%)	38.0	38.5	
≥200 (%)	0	7.7	
HbA_1c_ (%)	5.5 ± 0.5	5.8 ± 0.3	0.112
HbA1c (%)			0.691
5.7–6.4 (%)	62.0	54.0	
≥6.5 (%)	38.0	46.0	
Triglyceride (mg/dL)	90.8 (54.9–126.8)	90.8 (75.5–106.2)	0.500
TyG index	8.2 ± 0.6	8.4 ± 0.3	0.450
Total cholesterol (mg/dL)	208.6 ± 69.9	221.9 ± 50.5	0.583
HDL-cholesterol (mg/dL)	63.9 ± 21.9	77.5 ± 27.3	0.176
eGFR (ml/min/1.73 m^2^)	81.3 ± 7.8	77.8 ± 13.5	0.423
GOT (U/L)	30.8 ± 27.6	23.0 ± 7.4	0.332
GPT (U/L)	29.9 ± 25.6	24.5 ± 20.3	0.558
GA (%)	15.5 ± 2.4	14.4 ± 1.2	0.157
Insulin (mU/L)	5.1 (1.6–8.6)	6.9 (4.0–9.8)	0.398
HOMA-IR	1.2 (0.4–1.9)	1.7 (0.9–2.4)	0.290
β-cell function	65.5 (11.0–120.0)	74.3 (43.9–104.7)	0.761
QUICKI	0.34 ± 0.06	0.36 ± 0.03	0.036 *
U_ACR_ > 30 mg/g	7.7	7.7	1.000

Abbreviations are the same as Table 1. * *p* < 0.05.

**Table 9 healthcare-09-01010-t009:** Determinants of various parameters using multivariable linear regression analysis in control vs. spouses by age, sex and BMI match.

Parameter	Multivariable	
	Unstandardized Coefficient β (95% CI)	*p*
Fasting glucose		
Controls	Reference	
Relatives	−0.001 (−0.351,0.348)	0.994
TyG index		
Controls	Reference	
Relatives	−0.001 (−0.065,0.063)	0.975
β-cell function		
Controls	Reference	
Relatives	17.732 (−54.076,89.540)	0.606
QUICKI		
Controls	Reference	
Relatives	−0.045 (−0.097, 0.007)	0.083

Values expressed as unstandardized coefficient β and 95% confidence interval (CI). Abbreviations are the same as in Table 1. Adjusted for hypertension, coronary artery disease, body mass index, log triglyceride, total cholesterol, HDL-cholesterol, eGFR, GOT, GPT and U_ACR_ > 30 mg/g.

**Table 10 healthcare-09-01010-t010:** Description summarizing all the significant findings of the study (Control group as reference).

Parameter	Unmatched with Age, Sex and BMI		Matched with Age, Sex and BMI	
	Relatives	Spouse	Relatives	Spouse
Age (year)	55.8 ± 11.9	62.9 ± 9.5		
Chronic disease medication control (%)	53.9	52.0		
Hypertension (%)	25.5	27.0	18.0	
Body mass index (kg/m^2^)	25.4 ± 5.2	24.3 ± 5.4		
Fasting glucose (mg/dL)	105.6 ± 12.8	102.2 ± 10.8	104.5 ± 9.9	
Fasting glucose (100–125 mg/dL) (%)			66.7	
OGTT (mg/dL)	153.9 ± 44.1	153.8 ± 51.8	149.7 ± 39.3	
OGTT (140–199 mg/dL) (%)	39.8	40.5		
HbA1c (%)			5.8 ± 0.5	
HbA1c (5.7–6.4%) (%)	55.1	54.1		
eGFR (mL/min/1.73 m^2^)			91.5 ± 20.6	
QUICKI				0.36 ± 0.03
Determinants of fasting glucose using multivariable linear regression analysis (95% CI)	5.611 (1.282, 9.940)		6.600 (2.425, 10.774)	
Age (per 1 year) (95% CI)	0.208 (0.024, 0.391)			
Body mass index (per 1 kg/m^2^) (95% CI)	0.619 (0.251, 0.987)			
Determinants of TyG index using multivariable linear regression analysis (95% CI)	0.053 (0.013, 0.093)		0.064 (0.024, 0.104)	
Age (per 1 year) (95% CI)	0.002 (0.000, 0.004)			
Body mass index (per 1 kg/m^2^) (95% CI)	0.005 (0.002, 0.009)			
Triglycerides (log per 1 mg/dL) (95% CI)	2.356 (2.271, 2.442)			
Determinants of β-cell function using multivariable linear regression analysis (95% CI)	−43.083(−83.235, −2.931)			
GPT (per 1 U/L) (95% CI)	1.945 (0.329, 3.561)			
Determinants of QUICKI using multivariable linear regression analysis (95% CI)	−0.017 (−0.033, −0.002)	−0.023(−0.042, −0.004)		
Age (per 1 year) (95% CI)	0.001 (0.000, 0.001)	0.001 (0.000, 0.001)		
Body mass index (per 1 kg/m^2^) (95% CI)	−0.002 (−0.004, −0.001)	−0.002(−0.004, −0.001)		

Values expressed as unstandardized coefficient β and 95% confidence interval (CI). Abbreviations are the same as in Table 1.

## Data Availability

Not applicable.

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
