# Peer review of "Common Risk Factors in Relatives and Spouses of Patients with Type 2 Diabetes in Developing Prediabetes"

_healthcare, 2021, doi:10.3390/healthcare9081010_

Round 1
Reviewer 1 Report
Please see the attached review.

Author Response
In my opinion the manuscript “Common Risk Factors in Relatives and Spouses of Patients 2 with
Type 2 Diabetes in Developing Prediabetes” by Wei-Hao Hsu et al. has a potential to give the
readers important information regarding the risk factors for prediabetes, insulin resistance, and β
cell function impairment in relatives and spouses of patients with type 2 diabetes.
However, the Authors did not improve the methodology of the study and therefore we still cannot
formulate valuable conclusions from the study. My comments and suggestions from the previous
review for JPM have not been implemented:
“The control group and the relatives’ and spouses’ groups differ significantly regarding the age and
the female/male ratio. It is a crucial bias, especially in the field of analysis of metabolic state.”
“We cannot compare the groups of relatives and spouses in the 6th and 7th decade of life,
respectively, with the control group in the 5th decade (difference for the age p<0.001). The aim of
the study regards the metabolic state that may be highly related to age as well as sex. Therefore, the
age and the female/male ratio differences implicate that the results are not reliable, and we cannot
formulate valuable conclusions from the study.”
The Authors added a comment to the manuscript (Discussion) “Lastly, the small and limited study
population was enrolled in our study. Therefore, we were not able to match 3 study groups with age,
sex and BMI.”, which is not sufficient.
If I may suggest a way to make the paper reliable, I would put an effort for a better matching the
relatives’ group, which is larger (n=98), with the spouses’ group, even if it leads to a smaller
number of the relatives.
Response: We totally agree for your opinion and it's really applicable to large cohort study like randomized controlled trial. However, in our situation with a small study population group,with regards to the age, we are studying the spouse of our index type 2 diabetic patient, which is mostly 5th to 7th decade, nearly similar age with our index type 2 diabetic patient is all we could enrolled. And for the relatives , we included mostly siblings, father and mother , and excluded son and daughter of the our index type 2 diabetic patient, just to avoid the bias of greater difference of age in relatives and spouse as you mentioned. But ,55.8 ± 11.9 age of the relatives and 62.9 ± 9.5 age of the spouse , with the mean age difference of 7 years is already the smallest difference we could have. Regarding with the sex, there are a lot of factors that might influence the metabolic rate ,and sex is the least factor , therefore , in our opinion , even we do matching only for sex, it won't change our result that much. But reduction of the number of the study population will affect a lot.
Moreover, the control group is taken from a community screening programme, which I believe is an
epidemiological programme and has a large cohort of participants. Probably the Authors could look
again and find a better matched control group.
Response: The laboratory testing of community screening program in our country is quite simple which included only few data like fasting glucose, total cholesterol, triglycerides , serum creatinine and urinalysis. The rest of the data were additional and funded by ourselves. However, the most difficult part for these participants is the OGTT. To find a better matched control group may need a longer time for us to finish this study.Here , please accept our apology for this, but thank you so much for your very kind and nice suggestion. We will consider it in our future work. This statement was added in our future perspectives after the limitation in our manuscript.
Otherwise, although as I mentioned before, the topic is very interesting, the paper does not give us
the reliable conclusions.
Reviewer 2 Report
The article has improved since last submission. The authors managed to answer all I requested and can now be further process for publication.
Author Response
Thank you so much
Reviewer 3 Report
The paper by Wei-Hao Hsu et al on the common risk factors in relatives and spouses of patients with type 2 diabetes in developing prediabetes covers an interesting topic of correlations between being related to or living with someone with diabetes and the risk of developing the disease by him/herself. The results are clean-cut and clear, however it would be worth emphasising the results were p value was siginificant by bolding them or denoting them with an asterix in the table.
Furthemore, the introduction section would benefit from an addition of more geographically relevant data on prediabetes coming from the area where authors of the paper are based in. It would also be worth mentioning who were the first degree relatives (mothers/fathers/siblings?) and whether they were living together in the same household or somewhere else. It is known that there is a genetic predisposition to diabetes between relatives that might be deepened by environmental factors, providing relatives live nearby and have similar life habits.
Family structures are different in the East and in the West so that might be something to account for in the future publications.
Finally, a table or a more graphical description summarizing all the significant findings of the study might benefit the paper. Overall, I recommend it for publication after some revision
Author Response
The paper by Wei-Hao Hsu et al on the common risk factors in relatives and spouses of patients with type 2 diabetes in developing prediabetes covers an interesting topic of correlations between being related to or living with someone with diabetes and the risk of developing the disease by him/herself. The results are clean-cut and clear, however it would be worth emphasising the results were p value was siginificant by bolding them or denoting them with an asterix in the table.
Response: The significant p value was already denoted with asterisk in the table 1-5.
Furthemore, the introduction section would benefit from an addition of more geographically relevant data on prediabetes coming from the area where authors of the paper are based in.
Response:The relevant data on prediabetes in Taiwan was added in the introduction line 49-50.
It would also be worth mentioning who were the first degree relatives (mothers/fathers/siblings?) and whether they were living together in the same household or somewhere else. It is known that there is a genetic predisposition to diabetes between relatives that might be deepened by environmental factors, providing relatives live nearby and have similar life habits.
Family structures are different in the East and in the West so that might be something to account for in the future publications.
Response: The description of the first degree relatives was added now in Method section of study participants line 70-71.
Finally, a table or a more graphical description summarizing all the significant findings of the study might benefit the paper. Overall, I recommend it for publication after some revision.
Response: Table 6 was added at the end of conclusion to summarize our significant findings.
Reviewer 4 Report
This nice paper tries to identify prediabetes in family members of T2DM patients. I have the following remarks, though:
- Scores, such as the HOMA-index (e.g. in the case of very low beta cell function), are not always reliable, and the authors should comment on this.
- Were auto-antibodies evaluated to exclude LADA (which can resemble T2DM)?
- In families with multiple diabetes patients, the prevalence and incidence of MODY are much higher. Has this been taken into account?
- The authors should state what they propose to be done next (future perspectives).
Author Response
- Scores, such as the HOMA-index (e.g. in the case of very low beta cell function), are not always reliable, and the authors should comment on this.
- Response: The additional information on unreliability of HOMA was now stated at line 270-282.
- Were auto-antibodies evaluated to exclude LADA (which can resemble T2DM)?
- Response: Since our study population were prediabetes with strong family history ,auto-antibodies for diagnosing newly onset diabetes were not evaluated to exclude LADA .And this will be stated in out limitation.
- In families with multiple diabetes patients, the prevalence and incidence of MODY are much higher. Has this been taken into account?
- Response: Yes. But in our study population for relative were all with either or both father , mother or sibling with type 2 diabetes . Therefore, MODY has the probability but is less likely in our study population. MODY genetic testing will be stated in our limitation.
- The authors should state what they propose to be done next (future perspectives).
- Response: Thank you. Our future prospective is now stated after limitation at line 329-332.
Round 2
Reviewer 1 Report
Please see the attached review.

Author Response
as attachment.

This manuscript is a resubmission of an earlier submission. The following is a list of the peer review reports and author responses from that submission.
Round 1
Reviewer 1 Report
Thank you for the opportunity to review the manuscript “Prevalence and Risk Factors for Prediabetes in Relatives and Spouses of Patients with Type 2 Diabetes 2” by Wei-Hao Hsu et al.
The aim of the study was to assess the prevalence of prediabetes in relatives and spouses of patients with type 2 diabetes and the risk factors for prediabetes, insulin resistance, and β cell function impairment. It is an important public health issue as well as a part of personalized medicine. The study could be valuable as the population is homogenous and health behaviour variables together with metabolic state variables were included in the analysis. I appreciate the work of the Authors. However, in my opinion the research was not conducted correctly. The control group and the relatives’ and spouses’ groups differ significantly regarding the age and the female/male ratio. It is a crucial bias, especially in the field of analysis of metabolic state. As Authors commented in the Discussion (lines 185-189):
“Moreover, the relatives and spouses were older and had less physical activity compared to the healthy controls, and the relatives also had a higher BMI than the healthy controls. These findings may explain the high prevalence of prediabetes in the relatives and spouses, as previous studies have reported that old age, physical inactivity and being overweight are risk factors for prediabetes.”
We cannot compare the groups of relatives and spouses in the 6th and 7th decade of life, respectively, with the control group in the 5th decade (difference for the age p<0.001). The aim of the study regards the metabolic state that may be highly related to age as well as sex. Therefore, the age and the female/male ratio differences implicate that the results are not reliable, and we cannot formulate valuable conclusions from the study.
I would suggest including to the study groups patients that are comparable regarding the age and sex. Then I believe the Authors may achieve the reliable results and conclusions.
Reviewer 2 Report
Nice paper about insulin resistance and beta cell function in relatives and spouses of patients with diabetes compared to a control group. In order to compare other parameters beyond BMI and age, that could be interesting utilize a control group matched by at least this parameters.
Reviewer 3 Report
The article is not really well written as English language should be revised. The study has a good design. The article is logically divided into sections and subsections. There are several tables and figures of good quality. The methods are presented in sufficient detail and statistical analysis is well performed. The references cited not always relevant and adequate. The work has an average degree of novelty and of good interest to the readers.
In my opinion, this paper can be recommended for publication after major revision, as follows:
- Line 35: Reference 1 and 2 seems repetitive and I suggest choosing one of them.
- Line 45: references should be in bracket.
- Line 46: references should be put at the end of the sentence.
- Study participants and design should be rephrased better (how many patients refused? How many patients were lost at follow up? Did the enrolled patients sign a written consent? Please specify).
- Line 67: “with 7 diabetes mellitus” what is that number stand for?
- Oral glucose tolerance test paragraph: a better description of the test is mandatory.
- Line 79: “and 8 administration” what is that number stand for?
- Line 113: “,10 total cholesterol” what is that number stand for?
- Discussion: in table 1 is reported patients’ lifestyle. Was it evaluated in the multivariate analysis? If not, why? It seems strange that lifestyle do not independently associate with HOMA-IR, HOMA-B and QUICKI.
- Line 248-250: Increase in transaminases levels may also be explained by the extract of the following article https://doi.org/10.3390/pr9010135 “high plasma glucose levels induce toxicity and activate the apoptosis pathway in the liver”
- Insulin sensitivity is also affected by the amount of adipose tissue as extracted by the following article DOI: 10.3390/antiox10020270 to be added into discussion: “In fact, the relationship between insulin resistance and amount of visceral adipose tissue mass is directly proportional. Moreover, weight loss has been reported to improve insulin sensitivity thanks to the reduction of visceral adipose tissue mass. Hence, adipose tissue regulates insulin sensitivity in target tissues.
An important limit in this study that should be reported as well is the small sample size.